# OpenReview forum: "Enabling Robust In-Context Memory and Rapid Task Adaptation in Transformers with Hebbian and Gradient-Based Plasticity"
_TMLR — Rejected by TMLR_

### Review · Reviewer_DEah · 2025-11-03

**Summary Of Contributions:**

The authors investigate whether fast plasticity (either Hebbian or based on gradient descent over self-generated losses) improves the performance of transformers in memory tasks.

They perform various experiments and report results which seem to indicate an advantage in some circumstances, and attempt to explain these results.

**Audience:**

No

**Audience Explanation:**

See above.

**Broader Impact Concerns:**

No broader impact concerns.

**Claims And Evidence:**

No

**Claims Explanation:**

The main problem is that the experimental procedures are simply not explained. As a result, it is impossible to assess the results - in fact it's not even obvious exactly what question is being asked.

How exactly does the meta-training (and test) happen? E.g. for the copying task, do you show one sequence+delay, observe the response, and perform a gradient update?

Do you do that both for the plastic and the non-plastic networks?

Exactly how many gradient updates are performed before testing? How many times is each sequence in the dataset seen?

Most importantly, why on earth would you only use 50 sequences for the training dataset? Meta-training over just 50 episodes is unlikely to be sufficient. If you do more training over the same 50 sequences, eventually the network will simply learn the sequences themselves. What's the motivation here?


The same problem applies to all experiments, which are not explained. How exactly does the meta-training (and testing) happen? When exactly do you apply gradients, and to what? (The problem is particularly acute for Experiment 4.4, where it is not even clear what the task is).

Similarly, the actual networks themselves are insufficiently described, with multiple undefined symbols in the equations.

- In section 3.2: What is y-bar? Also: "5% computational overhead" - overhead in what, memory or CPU time? What is the added cost in both memory and CPU time?

- Equation 2: explain what the little circle means (I suppose it means "pointwise product" or something like that but it should be explained).

- Can you explain what equations 3, 4 do? What is eta-tilde?

- Equation 5: what is Wout? What is y-bar and eta-tilde? Why would concatenating these output amount to the same dimensionality as the network's outputs? Please explain!

Minor observations:

- The method of applying online gradient descent to a learned loss is reminiscent of Test-time-training or Titan (e.g. https://arxiv.org/abs/2504.05298, https://arxiv.org/abs/2501.00663 and references therein). Please explain similarities and differences of the gradient-based plasticity method described here with these papers.

- Throughout, citations are incorrectly typeset with parentheses in the wrong place. It appears that the authors confused the latex commands \cite / \citet / \citep.

- Literature: for learning synaptic rules, please cite Bengio et al. "On the optimization of a synaptic learning rule", 1997.

- I'm a bit confused about section 2.3. Currently, transformers do not have fast-weights / plasticity, therefore, ICL in current transformers cannot be of the "explicit" kind? Please clarify.

**Requested Changes:**

Please rewrite the paper so that the experiments are clearly explained, with a clear description of how meta-learning occurs, when gradient updates are applied (an how many), etc.

Also please resolve all undefined symbols in the equations (and explain the equations themselves).

Also, please explain why you use such small datasets.

---

### Review · Reviewer_1mVU · 2025-11-11

**Summary Of Contributions:**

The submission studies the impact of explicit parameter adaptation as a means to endow decoder-only transformers with plasticity, in contrast to in-context learning as a form of implicit adaptation. For this, the authors replicate the empirical study of Duan et al. (2023), originally applied to RNNs, on transformers. Their results highlight differences in the performance of Duan et al.'s (2023) Hebbian and gradient-based update rules on a variety of tasks.

######## Strengths ########

- The "related work" section does an excellent job of connecting various lines of research, in particular meta-learning, plasticity, and in-context learning in an insightful way. The bridges between these various threads is worth putting into words.
- The problem setting as explained through Sec 3.1 is quite interesting. The idea that we can treat a collection of ICL problems in an outer loop to learn a learning mechanism for doing ICL via parameter adaptation, using either a Hebbian update rule or a gradient-based update rule, is quite enticing. If it were possible to do this in a way that maintains all of the original transformer's capabilities, that has the potential to be incredibly impactful.
- The writing, up to section 3.1, is clear, concise, and thought-provoking. It does a good job of setting up what seems to be an exciting problem and guiding the reader through how it relates to existing work.

######## Weaknesses ########


- I have two "overall" concerns that I would hope to see discussed or at least acknowledged around Sec 3.1:
    1. I'm assuming that the "sample batch of sequences" step does the sampling within a particular domain (e.g., few-shot image classification). This will result in (static) transformer weights that indeed enable faster adaptation to in-domain sequences, but what effect should we expect on the transformer's behavior for other problems? I wonder about: implicit ICL to other domains, explicit/dense ICL to other domains, and general capabilities of the transformer outside of ICL.
    2. One of the key advantages of ICL, which has made it widely adopted, is that it requires no model access, no local memory, no memory overhead (beyond the context length increase, which the proposed method also incurs). The proposed approach requires full-finetuning (in the outer loop) + the fast weights and gradient overhead. What is the setup that the authors envision this being useful for, especially for closed models? Is the hope that maybe the model creators will perform a bunch of outer-loop updates to turn their models into great in-sequence learners (via fast weight updates) and then either release their weights for end users to do local fast weights adaptation or provide fast adaptation pipelines that run on the host's side?
    - Please note that I am not stating whether the proposed approach is/isn't valuable. I merely would like to see a discussion on these points to contextualize the contribution in light of what capabilities it may provide if widely adopted (or what gaps remain for future work in this direction to address).
- One of the advantages of ICL is precisely that it does not update parameters, thereby bypassing catastrophic forgetting. The "superior" gradient updates are only so for the target tasks, but not necessarily for the broader use of the transformers, I'd expect. Although I am curious how this relates to the authors' meta-learning, outer-loop approach, which presumably will be trained to improve adaptation to new contexts. If this is a very general thing that applies to "all" in-context tasks (so, if the outer loop samples across a broad range of diverse in-context tasks), then perhaps it would provide evidence that it continues to result in a sort of general-purpose transformer (i.e., one that does not forget)

**Additional Comments:**

The following points are provided as feedback to hopefully help better shape the submitted manuscript, but did not impact my recommendation in a major way.

Abstract
- The Abstract suggests that only the gradient-based plasticity is taken from Duan et al., but the Hebbian rule is, too. This should be made clear
- The submission is an *evaluation* paper to try to understand the relative merits of three adaptation techniques applied to a set of benchmarks. This isn't explicitly stated, and the authors instead claim that they "augment decoder-only...", which looks like overpromising.
- What are "short associations?"

Intro
- "the model's capacity to incorporate new evidence is limited to transient activations within self-attention, with no dedicated process for storing or consolidating information during inference." --  Given the massive contexts that current decoder-only architectures can manage, is it still fair to call the context "transient"? Conceptually, I agree with the authors' statement, but it may be worth considering a softer version of this claim.
- "Can explicit plasticity mechanisms improve a Transformer’s ability to adapt within a sequence?" -- At this point, I thought this would be equivalent to considering the "transient" context. I believe that, in the context of this submission, the distinction is the fact that there is an outer loop training the retains changes across sequences. This is not made explicit.
- What is a "fast-weight component"? Is it a separate module of weights (e.g., LoRA) or a fast update rule for the main transformer weights? This only becomes clear much later.

Sec 3.2
- Does "self-attention matrices" here mean key, query, and value projection matrices? The self-attention matrix itself is not a learned parameter, which is why I'm confused.

Sec 3.3
- The notation for Eq. 4, taken directly from Duan et al., is not clear, because it is not clear that the input to Concat is a set. Adding curly braces around the input to Concat would work better.

Sec 3.4
- For consistency, if the authors choose to adopt the set notation for Eq. 4 (as I suggest above), they should also use it for Eq. 5.
- I also encourage the authors to explicitly define $\delta$ in a new Eq. 8, analogous to Eq. 4.
- While this time the authors did cite Duan et al. in Sec 3.3, they failed to acknowledge the source for the approach in Sec. 3.4, which is also directly taken from Duan et al.

Typos/style/grammar
- Sec 4.1: "Every runner" -> "Every run"?

**Audience:**

Yes

**Audience Explanation:**

Understanding the dynamics of in-context learning and contrasting it to explicit weight updates is of broad interest to the community.

**Broader Impact Concerns:**

None.

**Claims And Evidence:**

No

**Claims Explanation:**

- The contributions, listed at the end of Sec 1, are significantly over-stated
    1. A framework for synaptic plasticity into autoregressive transformers. How is this framework different from the one of Duan et al.? From a quick read of the original reference, it seems to me like this submission is a direct application of their framework to transformers, with no modification. In fact, while the authors cite Duan et al. in multiple places and describe the proposed approach as a parallel to Duan et al., I still don't get a clear picture of which elements of this submission are distinct in terms of the technical learning procedure. Is it only the application of the same approaches to transformers? Is there any technical difference? It would be helpful if Sec 2.1 provided a more explicit list of technical advances w.r.t. Duan et al. (if they exist) or statement that precisely states that the technical approach is identical and the contribution is the adaptation/application to transformers + empirical analysis compared to implicit ICL.
    2. A systematic comparison of Hebbian and gradient-based plasticity. This appears to be the primary contribution, and the Abstract/Sec 1 should be framed around it, if that is the case.
    3. Mechanistic insights into how explicit plasticity interacts with self-attention dynamics. It is unclear to me how Figure 2 in the appendix, which is the only result that is highlighted as "mechanistic" relates to self-attention dynamics in any way. The entire presentation and discussion of these insights covers three lines of text (second paragraph of Sec 4.7). Without a deeper dive into these mechanistic insights, I would be hesitant to treat these as significant insights.
- A few details are insufficiently explained in Sec 3.1:
    - What is the "overall meta-loss"? Does it depend on the choice of Hebbian vs gradient update?
    - What is d_model? This is the first time it is introduced. Presumably these are the input=output dimensions of the FFNs? It is useful to know that this only introduces a 5% compute overhead, but what is the corresponding (percent) memory overhead?
- I found Sec 4 largely unclear and not as carefully connected to the remainder of the paper, in particular w.r.t. its relation to ICL on LLMs
    - Sec 4.1
        - It is unclear to me what a non-plastic baseline would train. If the network is not plastic, aren't all parameters fixed and therefore there is no training? Is this supposed to be the baseline "implicit" ICL? If so, what are the "identical hyperparameters" used to train it, if ICL does not use explicit training?
        - It is also unclear to me what a "qualitative ranking of rules" is. Should this not be simple a "ranking of rules"?
    - Sec 4.2
        - Table 1 should be referenced in text so the reader knows where to look when reading this section.
            - Are the "loss" values in this table directly comparable to those in Duan et al.?
            - If so, all three methods evaluated on the transformer exhibit drastically worse evaluation performance than the RNNs from Duan et al.?
        - Figure 2 in the Appendices should be referenced when discussing the neuromodulation and plastic-weight norms. [Perhaps the authors did not mean to include statements about neuromodulation and plastic-weight norms here, since the statement is repeated in Sec 4.7?]
            - This Figure generates a lot of confusion
            - What is an epoch here? A pass through the whole sequence? I find it very odd to think about training for multiple epochs given the motivation of the authors of learning within a sequence.
            - I'm also confused about why the neuromodulation/plastic weight norms are shown as one point per epoch. Wouldn't we get a different value for each time step in the sequence?
        - I'm also generally confused about the training setting for this task.
            - The abstract and Introduction discuss the setting as applicable to LLMs and comparable to in-context learning. Is the transformer used in this section an LLM? Is it a pretrained transformer or trained from scratch?
            - What constitutes the "outer loop" in this setting? Is the transformer seeing many "copying task" instances and then being evaluated for its ability to rapidly learn one new copying task?
            - It seems that some of these details are provided in Appendix A, but this is never referenced nor are sufficient details provided
    - The comments from Sec 4.2 apply equally to 4.3-4.5
    - Sec 4.6
        - Why is it the case that RNNs perform that much better than transformers on these problems?
    - Sec 4.7
        - What is the significance of the neuromodulation ablation? Is it not trivial that setting the effective learning rate to 0 will effectively do no learning?
    - Sec 4.8
        - "Extending the copying task to 8-layer models" -- This is the first time anything is mentioned about the model architecture, since the reader was never referred to Appendix A.
        - Where do we see the results of the divergence at 3000 steps? Also, what does 3000 steps mean? Is is 3000 outer loops or inner loops?
        - "recall below baseline" -- what below baseline?
        - How could we determine that a "practical regime therefore lies around 4 layers" if we do not see an evaluation at 4 layers but only 2 and 8 layers?

**Requested Changes:**

Summarizing the concerns highlighted in previous boxes, here are some requested changes:

Critical to recommend acceptance:
1. Clarify the contributions made by this submission without overpromising — i.e., there is no novel adaptation framework, and there are minimal "mechanistic" insights; the main contribution appears to be the empirical evaluation on transformers
2. Improve the description of the method as explained in Sec 3.1
3. Clarify the experimental setting and ensure that it is sufficiently connected to the motivation throughout Sec 1 and 2 on LLMs and ICL

Recommended to strengthen the work:
1. Include discussion of the effect of outer-loop training within a domain on the capabilities of a pretrained LLM
2. Discuss the possible uses of the proposed adaptation setting in the space of LLMs
3. Discuss the potential effects of catastrophic forgetting on a pretrained LLM after outer-loop training and contrast them to ICL

---

### Review · Reviewer_qGxQ · 2025-11-16

**Summary Of Contributions:**

Duan et al (2023) examines "Hebbian and Gradient-based Plasticity Enables Robust Memory and Rapid Learning in RNNs". This paper builds upon it and expands the analysis to Transformers. In particular, it expands a regular decoder-only Transformer architecture with "fast" weights, which are adapter weighst on the FNNs that are trained during autoregressive decoding, either using (1) Hebbian learning or (2) using a gradient-based approach. These approaches are compared to a regular model without these fast weights in four settings (copy, cue-reward association, one-shot image classification, and few-shot regression). Performance gains are modest compared to non-plastic baselines while significantly underperforming plastic RNNs.

**Audience:**

No

**Audience Explanation:**

As is, and especially when comparing to Duan et al (2023, https://arxiv.org/abs/2302.03235), I'm not sure whether this submission will be of interest to the community. I don't think the experiments conclusively show that either approach works better than none. Esp compared to Duan, this submission lacks similar plots and detailed analyses.

**Claims And Evidence:**

No

**Claims Explanation:**

The paper is clearly written and overall effectively motivates its research questions.

The contributions, however, seem slightly overstated and thus not evidenced by the experiments.

1. "Systematic comparison on various benchmarks" when it is only compares on 4 toy settings and classification on CIFAR-FS/Omniglot (the most serious experiments imo)
2. "Mechanistic insights" boils down to reporting a singular mean value for the global neuromodulation factor per approach per experiment setting.

Re "general framework": the paper only presents results for a 2-layer decoder Transformer (acc to §A.1) and only adapts the FFN. It does not explain why attention matrices are not considered for adaptation. Further, as mentioned in §4.8, the approaches diverge for deeper Transformer architectures (with 8 layers). This somewhat contradicts the "robust" adaptation mentioned in the title.

Regarding the reported experiment results, there is no statistical significance testing despite using only three seeds, and the performance improvements are marginal, often falling within the error bars. The latter, in particular, when it comes to the comparison between Hebian and gradient-based plasticity.

Finally, the plastic Transformers substantially underperform plastic RNNs on identical tasks. Usually, Transformers perform better than RNNs and are easier to train.

**Requested Changes:**

https://arxiv.org/abs/2302.03235 has many plots that I would have expected to see in this submission as well (either main or appendix). Of particular importance are plots that visualize the global modulation factor across different tasks and across task length. Only reporting a single mean per task is not sufficient to provide mechanistic insights. Would it be possilbe to add these and also loss curves for the experiments and given the similarities to Duan follow Duan more closely in the plots and analysis as well (and compare to it in more detail in the experiments section too)

It would be useful to conduct significance testing and use more than three seeds to ensure that the experiment results are convincing. I am not convinced by the statements that compare Hebian and gradient-based plasticity. Could this be done?

Further, in the RNN setting, both Hebian and gradient-based plasticity outperforms the regular model. Here, it is the opposite. It would be good to investigate this.

Overall, I would also suggest updating the title and contributions to tone them down.

### Minor

Citations are using the wrong format.

In §3.4, the global modulation $\eta$ is not defined.

§4.2 in the interpretation it says: Gradient-based plasticity achieves the lowest loss but actually it is Hebbian (again if only look at the mean and do not look at the overlap).

§A.1: §3 does not mention that the model is a /two-layer/ decoder Transformer.

---

### Comment · Reviewer_DEah · 2025-11-10
**Review of updated paper**

The paper has been improved, but remains problematic.

**Main problem:**

- The main problem is that the differences shown in Figure 1 are extremely small, and it is not at all obvious whether the results are significant. Please provide some quantitative evidence of significance, i.e. some p-values (you may need to increase the number of seeds).

**Other problems:**

- Frustratingly, the tasks are still not explained. "CIFAR-FS and Omniglot episodes use 5-way, 1-shot support sets with 15 query images per class" - what does that mean? How is "Query loss" computed? Etc. Please provide a sufficiently detailed description and explanation of each task that would allow a reader to reproduce them.

- In Section 4.1, please specify exactly all the parameters that are updated at each step in the inner loop, and all those that are updated in the outer loop (e.g. what about Wout?).

- In 3.3 we read that eta(t) is "learned". The rest of the paper strongly implies that it is not learned, but that eta-tilde (from which eta is computed through the fixed formula Equation 3) is an output of the network (like in differentiable plasticity). Please clarify.

- In the non-plastic version (rule=none), does anything get updated in the outer loop? The whole experiment suggests that the base weights would be updated in the outer loop also for the non-plastic version. Is it correct?

- Does training / meta-training also affect the self-attention weights, or only the feedforward MLP weights (for both the plastic and non-plastic version)?

- Apparently the total meta-training times are extremely short, with a few hundred episodes/sequences at most (typical meta-training experiments often involve many thousands of episodes). Do the results change appreciably if you double the number of epochs?

- Section 4.1: Is each epoch made of completely new sequences (so each sequence is seen only once during the whole training)? Or is each sequence seen "once per epoch" (so each sequence is seen E times)? The paragraph may be understood both ways. Please clarify.

- Section 6, Compute: the compute times are reported to be a few minute, but just below the experiments are reported to take 25 hours. What is the source of the difference? Also, is there a difference in compute time between the plastic and non-plastic version?

- Many references are still messed up. The authors may consider using the following Latex command: \renewcommand{\cite}{\citep} (and then explicitly use \citet where necessary).

---

### Decision · Action_Editor_3TPY · 2025-12-17

**Recommendation:** Reject

**Audience:**

Yes

**Audience Explanation:**

On the whole, several reviewers and I agree that even if this paper does not dramatically advance the field, aspects of would be interesting to a subset of TMLRs audience since the topic is important and timely.

**Claims And Evidence:**

No

**Claims Explanation:**

All three reviewers found that the paper made much stronger claims than it could support, failed to include adequate evidence for the claims that it could potentially support (e.g. statistical tests that the improvements were meaningful, or supporting analyses that would bolster the claims such as modulation across tasks, loss curves, etc.), and failed to clearly explain what exactly was done and why it should matter. Thus, the experiments would need to be thoroughly reworked, and the paper would need to be thoroughly rewritten to merit acceptance.

**Resubmission Of Major Revision:**

The authors may consider submitting a major revision at a later time.